# Potassium Nitrate Treatment Is Associated with Modulation of Seed Water Uptake, Antioxidative Metabolism and Phytohormone Levels of Pea Seedlings

José A. Hernández *, Pedro Díaz-Vivancos *, José Ramón Acosta-Motos and Gregorio Barba-Espín *

Group of Fruit Trees Biotechnology, Department of Plant Breeding, CEBAS-CSIC,
Campus Universitario de Espinardo, 30100 Murcia, Spain; jacosta@cebas.csic.es
* Correspondence: jahernan@cebas.csic.es (J.A.H.); pdv@cebas.csic.es (P.D.-V.); gbespin@cebas.csic.es (G.B.-E.)

**Abstract:** (1) Background: Seed treatment with potassium nitrate ($KNO_3$) has been associated with dormancy breaking, improved germination and enhanced seedling growth and uniformity in a variety of plant species. However, the $KNO_3$ effect seems to be dependent on plant species and treatment conditions. (2) Methods: We describe the effect of incubation of dry pea seeds with different $KNO_3$ concentration on water uptake kinetic, early seedling growth, antioxidant metabolism and hormone profile in pea seedlings. (3) Results: Low (0.25 mM) $KNO_3$ levels increased seedling water uptake and growth, whereas high (40 mM) levels decreased seedling growth. $KNO_3$ treatment differentially affected the antioxidant defences. Low $KNO_3$ levels maintained the activity of antioxidant enzymes, while high levels reduced the activity of $H_2O_2$-scavenging enzymes. $KNO_3$ induced a progressive decline in ascorbate levels and reduced (GSH) and oxidised (GSSG) glutathione. Low $KNO_3$ levels strongly increased $GA_1$ and decreased ABA in both seedlings and cotyledons, resulting in a decline in the ABA/GAs ratio. (4) Conclusions: Pea seed treatment with a low $KNO_3$ level promoted early seedling growth. In this process, an interaction among $KNO_3$, antioxidant defences and ABA/GAs ratio is proposed.

**Keywords:** ABA; antioxidant defences; GAs; nitrate; seed germination; seedling growth; water uptake

## 1. Introduction

Seed germination is the most critical stage in crop establishment, determining crop production [1]. Numerous methods have been used to promote seed germination and seedling establishment under normal and stressful conditions. In this sense, seed priming, defined as a pre-sowing treatment which involves controlled hydration of seeds during the first stage of germination, has been widely applied to improve the germination rate and seedling growth under different stress conditions [2–5]. Likewise, seed chemical treatment during imbibition has been successfully applied for both fundamental research purposes and for the stimulation of seed germination and seedling vigour [6–8].

Potassium nitrate ($KNO_3$) has been demonstrated to break seed dormancy, promote seed germination and enhance growth uniformity in a variety of plant species, including tomato, maize, Arabidopsis and pea [9–12]. However, the mechanism by which $KNO_3$ improves seed germination and seedling establishment remains unclear. The priming effect of $KNO_3$ seems to be dependent on the $KNO_3$ concentration and application method. For example, improved seedling establishment was observed when seeds of *Paspalum vaginatum* (cv. Sea-Spray) were imbibed in 20–50 mM $KNO_3$ for three days [13]. Conversely, other authors reported that water-imbibed pea seeds (*Pisum sativum* cv. Lincoln) displayed reduced germination and seedling growth when incubated for three days in the presence of 30 or 40 mM $KNO_3$, whereas 10 mM $KNO_3$ stimulated early seedling growth. However, when the pea seeds were imbibed with $KNO_3$ (40 mM) for 24 h and then incubated in distilled water, seedling fresh weight was enhanced, whereas germination rate remained

unchanged [12]. Other authors [14] suggested that nitrates may enhance seed germination and early seedling growth due to the dual role of N as a plant essential element for growth and as a signalling molecule.

The effects of $KNO_3$ on plant hormones' regulation are unclear, with most studies focusing on abscisic acid (ABA) levels. $KNO_3$ has been reported to positively affect seed germination by modulating ABA metabolism or ABA signalling in developing seeds [15,16]. Gibberellins (GAs) are also important plant hormones for numerous physiological plant processes, including seed germination [17]. Other authors [18] reported that activation of ABA catabolism and GAs biosynthesis is required for seed germination. Low nitrate concentration decreases ABA content, leading to the induction of the *CYP707A2* gene, which encodes an ABA 8′-hydroxylase involved in ABA catabolism [15,19]. In addition, the up-regulation of the *CYP707A2* gene precedes the induction of the *GA3ox2* gene, related to GA biosynthesis [20]. Vidal et al. [12] described that exogenous $KNO_3$ enhanced $GA_4$ content and reduced ABA levels, resulting in a decrease in the ABA/GAs ratio. This effect was reversed by the action of the nitric oxide (NO)-scavenger 2-4-carboxyphenyl-4, 4, 5, 5-tetramethylimidazoline-1-oxyl-3-oxide (cPTIO), suggesting an interplay between $KNO_3$ and NO metabolism [21]. This in turn would indicate the role of NO in seed germination and seed dormancy breaking [9,22].

In addition, exogenous $KNO_3$ has been reported to lead to an increase in the expression of genes involved in N and C metabolism, as well as in energy production [15]. A role for nitrate reductase induction, related to N assimilation, and the antioxidative metabolism has also been described [11,12]. In that regard, the application of $KNO_3$ has been reported to increase antioxidant enzymes activity (superoxide dismutase (SOD), catalase (CAT), peroxidase (POX) and ascorbate oxidase (AOX)) in seedlings from different plant species [11,12]. During imbibition, seeds take an increasing amount of oxygen, causing the accumulation of reactive oxygen species (ROS) and a shift in the redox state [23]. Although ROS were originally considered as toxic by-products, different studies revealed that they are used by most organisms as key signalling molecules [24]. In addition, it was recently described that a basal level of ROS is required to support life [25]. In fact, it was recently suggested that the type of oxidative modification integrated into different oxidative signalling pathways regulates many crucial aspects of plant biology [26]. Several works have described the implication of ROS and antioxidant metabolism in the germination process. The scientific literature contains plenty of information concerning the beneficial effects of ROS on germination and seedling growth processes [6–8,27–31]. In these works, the authors showed that a controlled ROS generation during seed imbibition may have a signalling function during germination as well as during the dormancy release process. In contrast, uncontrolled ROS accumulation could delay or even inhibit seed germination [31]. In addition, an interplay ROS/plant hormone has been described during the seed germination process [6–8,28,29].

In the present work, we attempted to explain if direct nitrate addition during imbibition may promote pea seed germination and seedling establishment and vigour. For this purpose, we treated dry pea seeds with different $KNO_3$ concentrations to study its effect on water uptake kinetic, early seedling growth and antioxidant metabolism-related enzyme activity in pea seedlings. In addition, the effects of $KNO_3$ on ABA and GAs levels in pea seedlings and cotyledons were also addressed.

## 2. Materials and Methods

### 2.1. Plant Material, Culture Conditions, Growth Measurements and Sampling

Pea (*Pisum sativum* cv. Lincoln) seeds were obtained from Ramiro Arnedo S.A, Murcia, Spain. In the first experiment, individual dry seeds were placed inside 3.5 cm diameter plastic cups, onto two discs of filtered paper moistened with 1 mL $KNO_3$ (0, 0.25, 0.5, 1, 5, 10, 20, 30, 40 or 80 mM). Twenty-five seeds per treatment were arranged in trays and placed inside plastic bags, containing some small holes, to avoid water evaporation, and incubated in darkness at 25 °C in an incubator (MIR-153, Sanyo, Osaka, Japan). The germination percentage and water absorption rate (µL $H_2O$/g dry weight (DW), calculated

as the difference between the final and the initial weights, divided by the initial weight) were registered daily during $KNO_3$ treatment from days 0 to 4.

Subsequently, and based on the results of the preliminary assays, dry seeds were placed on 15 cm diameter Petri dishes, onto two discs of filtered paper moistened with 7 mL $KNO_3$ (0, 0.25 or 40 mM). Three to six Petri dishes per treatment containing 20 seeds per plate (biological replicate) were arranged and incubated at 25 °C for four days in an incubator (MIR 153, Sanyo) in darkness. Subsequently, seedling growth (length and mass) was recorded. Seedlings were separated from cotyledons and both were frozen in liquid nitrogen and stored at −80 °C for further analysis.

### 2.2. Enzyme Extraction and Assays

All operations were carried out at 4 °C. Four-day-old pea seedlings were homogenised and prepared for enzymatic analyses as described [12]. The activity of the antioxidant enzymes (ascorbate (ASC)-glutathione (GSH) cycle enzymes, SOD, POX, CAT and ascorbate oxidase (AOX)) was measured as previously reported [6,32]. The protein concentration was calculated according to Bradford [33]. The analyses were performed in a UV/Vis V-630 Bio spectrophotometer (Jasco, Tokyo, Japan).

### 2.3. Ascorbate and Glutathione Analyses

Frozen pea seedlings were ground into a fine powder in the presence of liquid nitrogen. Then, a 1M $HClO_4$ solution containing 1 mM EDTA and 1% (*w/v*) polyvinylpyrrolidone phosphate was added (1/3, *w/v*). The resulting extract was centrifuged at 12,000× *g* for 10 min at 4 °C. The pH of the supernatant was adjusted to 5.5–6 with 5 M $K_2CO_3$. The new mixture was centrifuged at 12,000× *g* for 1 min to eliminate the precipitate of $KClO_4$ formed. The resulting supernatant was used to quantify the oxidised and reduced ascorbate and glutathione forms [32].

### 2.4. Analysis of Plant Hormones

The analysis of ABA and GAs was performed in seedlings and cotyledons. Frozen samples were first lyophilised and then ground into a fine powder in a mortar. One 30-mg aliquot of each sample was sent to the Plant Hormones Quantification Service of the IBMCP (CSIC, Valencia) and quantified as described [34].

### 2.5. Statistical Analyses

Analyses of germination percentage, water uptake and seedling growth were performed on at least 20 biological replicates (individual seedlings). The remaining analysis was on three to six biological replicates (the pool of seedlings from a plate). Data were expressed as the mean ± SE. The data were analysed by one-way ANOVA followed by a Tukey's multiple range test ($p \leq 0.05$) using the SPSS 26 software (IBM SPSS Statistics, Chicago, IL, USA). All experiments were repeated independently at least twice with similar results.

### 3. Results

In a preliminary experiment, we assayed the effect of different $KNO_3$ concentrations on the water uptake rate and germination rate of pea seeds. In general, higher $KNO_3$ concentrations (10–80 mM) delayed seed germination, although at days 3 and 4 the germination rate was similar to that of control seeds (Supplemental Table S1a). Lower $KNO_3$ concentrations (0.25–5 mM) showed similar germination rates when compared to the control treatment at every day of treatment (Supplemental Table S1b). Taking into account that the highest seedling fresh and dry weights were registered with 0.25 mM $KNO_3$, this concentration, together with 40 mM $KNO_3$ as a representative of a high concentration, were used for further experiments.

Figure 1 shows the seedling water uptake for the control, 0.25 mM and 40 mM $KNO_3$. The 40 mM $KNO_3$ treatment did not alter seedling water uptake compared to control seeds,

whereas 0.25 mM KNO$_3$ increased it significantly with respect to the control at days 1 and 3.

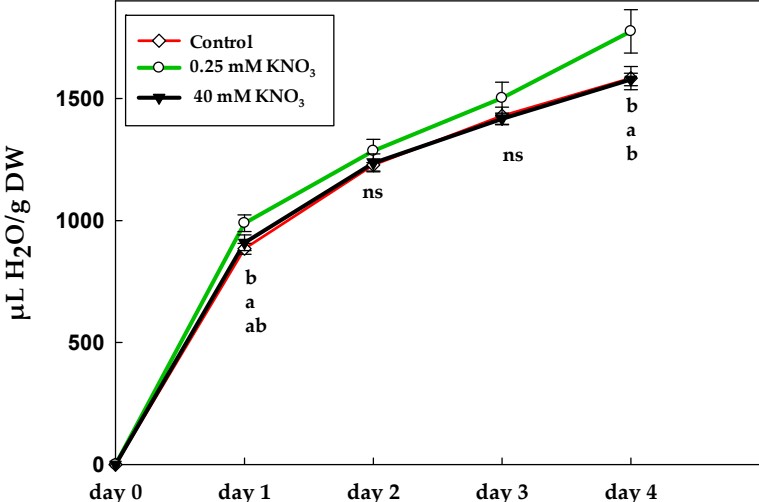

**Figure 1.** Effect of KNO$_3$ on the rate of water uptake of pea seedlings during 4 days of incubation. Each seed was weighed before being subjected to any treatment (initial dry weight (DW), day 0), and was then deposited in the bottom of a 3.5 cm diameter plastic cup, onto two discs of filter paper moistened with 0 (control), 0.25 and 40 mM KNO$_3$. The seeds were weighed daily and the rate of water absorption calculated as μL H$_2$O/g DW. Different letters (ordered from the top to the bottom for control, 0.25 mM and 40 mM KNO$_3$, respectively) within the same day indicate significant differences according to Tukey's test ($p \leq 0.05$).

A different effect for 0.25 and 40 mM KNO$_3$ was observed in the plant growth parameters (Figure 2). In general, in pea seedlings, the low KNO$_3$ concentration increased both FW and length by more than 15%, whereas 40 mM KNO$_3$ decreased the length and FW by 15% and 25%, respectively (Figure 2).

According to the ANOVA analysis, the KNO$_3$ treatments significantly affected all the enzymes of the ASC-GSH cycle as well as POX activity, measured at day 4 of treatment. At 40 mM KNO$_3$, a near 2-fold decrease in APX and POX activity was observed (Table 1). Conversely, with the same treatment, DHAR and GR significantly increased by about 60%. An increase in MDHAR, DHAR and GR (ascorbate and glutathione recycling activities) was also observed for the 0.25 mM KNO$_3$ treatment (Table 1). Moreover, POX activity significantly increased (near 2.5-fold) in 0.25 mM KNO$_3$ with respect to the 40 mM KNO$_3$ treatment, though the increase with respect to the control (27%) was non-significant (Table 1).

**Table 1.** Effect of KNO$_3$ on the activity of different antioxidant enzymes on four-day-old pea seedlings. Ascorbate oxidase (AOX), ascorbate peroxidase (APX), monodehydreascorbate reductase (MDHAR), dehydreascorbate reductase (DHAR), glutathione reductase (GR) and peroxidase (POX) activity are expressed in nmol $\times$ min$^{-1}$ $\times$ mg$^{-1}$ protein. Superoxide dismutase (SOD) activity is expressed in Units (U) $\times$ mg$^{-1}$ protein.

| KNO$_3$ (mM) | AOX | APX | MDHAR | DHAR | GR | POX | SOD |
|---|---|---|---|---|---|---|---|
| 0 | 60.5 ± 4.1 | 39.9 ± 4.2 a | 198 ± 12.9 b | 0.52 ± 0.05 b | 50.8 ± 3.0 b | 1383 ± 172 ab | 34.8 ± 2.5 |
| 0.25 | 51.9 ± 1.7 | 34.8 ± 2.2 a | 316 ± 15.9 a | 0.80 ± 0.08 a | 85.8 ± 3.1 a | 1765 ± 243 a | 37.5 ± 3.8 |
| 40 | 53.1 ± 3.8 | 18.2 ± 2.2 b | 165 ± 19.9 b | 0.85 ± 0.04 a | 80.6 ± 3.9 a | 713 ± 43 b | 27.7 ± 2.8 |
| [a] F | 2.39 ns | 13.7 ** | 23.4 *** | 23.4 *** | 31.7 *** | 8.46 ** | 2.71 ns |

[a] Significant F values from one-way ANOVA are denoted at 99.9% (***) or 99% (**) levels of probability (ns, not significant). Data represent the mean ± SE from at least six measurements. Different letters indicate significant differences according to Tukey's test ($p \leq 0.05$).

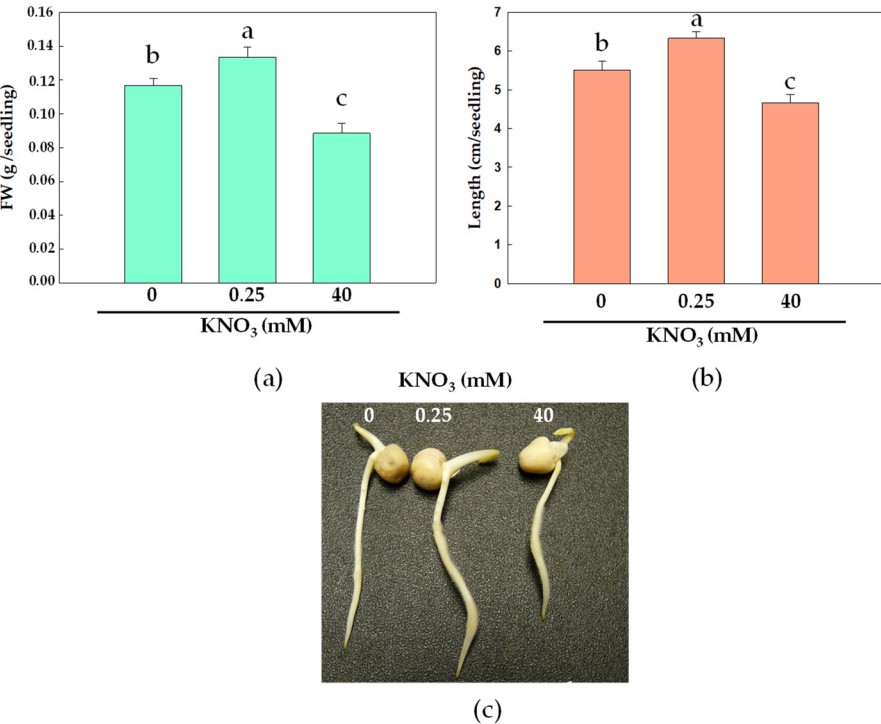

**Figure 2.** Effect of $KNO_3$ on the (**a**) fresh weight (FW) and (**b**) length of pea seedlings after four days of incubation. (**c**) Picture showing the visual effects of $KNO_3$ on seedling growth. Data represent the mean $\pm$ SE from at least 30 measurements. Different letters indicate significant differences according to Tukey's test ($p \leq 0.05$).

The $KNO_3$ treatments also affected the ascorbate and glutathione content (Table 2). In that sense, $KNO_3$ strongly decreased the reduced ascorbate (ASC) content in a concentration-dependent manner (Table 2). The oxidised ascorbate (DHA) was only detected in those samples treated with the highest $KNO_3$ level (Table 2). At 0.25 mM $KNO_3$, a decrease in both reduced (GSH) and oxidised (GSSG) glutathione levels, and thus in the total concentration of glutathione, was observed. However, an increase in the redox state of the glutathione pool was observed (Table 2). In contrast, at 40 mM $KNO_3$, GSH and total glutathione levels were not affected, though a decrease in GSSG was observed and, consequently, an increase in the redox state of the glutathione pool was registered (Table 2).

**Table 2.** Effect of $KNO_3$ on ascorbate and glutathione concentration in pea seedlings. Reduced (ASC) and oxidised (DHA) ascorbate and reduced (GSH), oxidised (GSSG) and total glutathione are expressed in nmol $g^{-1}$ fresh weight (FW). (nd: not detected.)

| $KNO_3$ (mM) | ASC | DHA | GSH | GSSG | GSH/ GSH + GSSG | Total Glutathione |
|---|---|---|---|---|---|---|
| 0 | $1401 \pm 73$ a | nd | $390 \pm 19$ a | $20.16 \pm 3.12$ a | 0.951 | $425 \pm 32$ a |
| 0.25 | $535 \pm 34$ b | nd | $263 \pm 14$ b | $9.27 \pm 2.06$ c | 0.966 | $281 \pm 11$ b |
| 40 | $375 \pm 85$ b | $7.60 \pm 4.39$ | $447 \pm 26$ a | $14.10 \pm 1.81$ b | 0.969 | $452 \pm 29$ a |
| [a] F | 60.9 *** | | 32.9 *** | 21.7 *** | | 15.2 *** |

[a] Significant F values from one-way ANOVA are denoted at 99.9% (***) levels of probability. Data represent the mean $\pm$ SE from at least six measurements. Different letters indicate significant differences according to Tukey's test ($p \leq 0.05$).

Potassium nitrate affected the ABA and GAs levels in both the seedlings and the cotyledons. $GA_4$ was detected in both tissues, while $GA_1$ was only detected in the seedlings. In seedlings treated with 0.25 mM $KNO_3$, a 6-fold increase in $GA_1$ as well as a slight decrease in $GA_4$ was observed (Figure 3a). In cotyledons, the effect of $KNO_3$ on $GA_4$ levels was not statistically significant (Figure 3b).

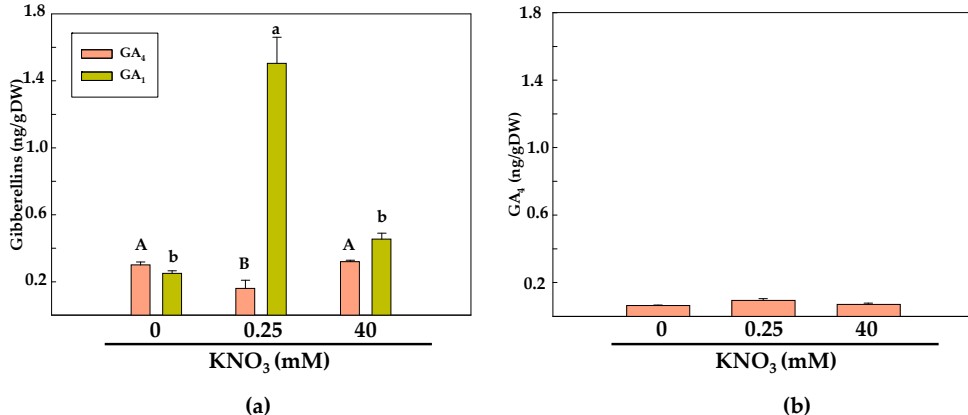

**Figure 3.** Effect of KNO$_3$ on gibberellins (GAs) levels in (**a**) pea seedlings and (**b**) cotyledons after four days of incubation. Data represent the mean ± SE from at least three measurements. Different letters (uppercase and lowercase letters for GA$_4$ and GA$_1$, respectively) indicate significant differences according to Tukey's test ($p \leq 0.05$). (DW, dry weight.)

The effect on ABA content varied depending on KNO$_3$ concentration and the type of tissue (seedling or cotyledon). In seedlings, at 0.25 mM KNO$_3$, a 54% decrease in the ABA level was observed, while 40 mM KNO$_3$ concentration produced a 1.7-fold increase in ABA, compared to control seedlings (Figure 4a). ABA levels in the cotyledons were six times lower than in seedlings under control conditions. At 0.25 mM KNO$_3$, ABA levels declined in the cotyledon, though no effect was observed at 40 mM KNO$_3$ (Figure 4a). As a consequence of the KNO$_3$-induced changes in ABA and GAs in pea seedlings and cotyledons, an important decrease in the ABA/total GAs ratio occurred in the cotyledons and, especially, in the seedlings (Figure 4b).

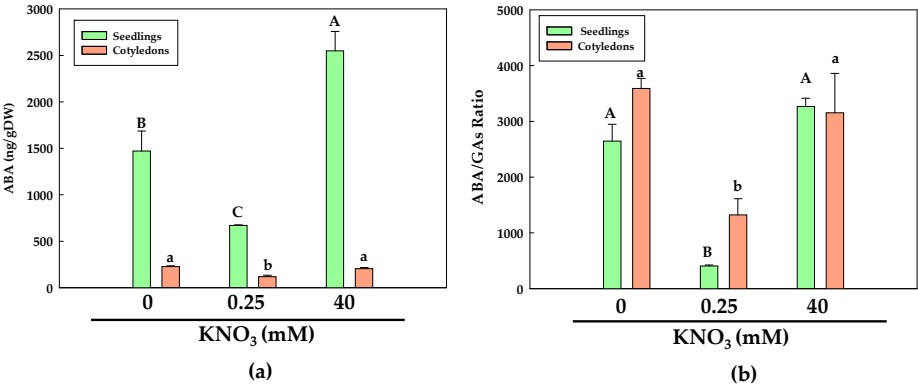

**Figure 4.** Effect of KNO$_3$ on (**a**) abscisic acid (ABA) levels and (**b**) ABA/GAs ratio in pea seedlings and cotyledons after four days of incubation. Data represent the mean ± SE from at least three measurements. Different letters (uppercase and lowercase letters for seedlings and cotyledons, respectively) indicate significant differences according to Tukey's test ($p \leq 0.05$). (DW, dry weight).

## 4. Discussion

Nitrogen is a macronutrient that may be a limiting factor for plant growth. At low concentrations, nitrate can stimulate seed germination in a variety of different plant species [14]. The mechanism of action of KNO$_3$ on the improvement of seed germination and/or early growth is far from being completely understood. The complexity about nitrate effects on seed germination and early seedling growth could be due to its dual role as a nutrient and a signalling molecule [14]. Nitrate stimulation of seed germination is often associated with plant species whose seeds require light for germination [35,36].

Potassium nitrate has been reported to improve pea seed germination and plant performance, but this effect is dependent on the concentration and the mode of application [1–4].

In the present work, we applied $KNO_3$ directly to dry seeds; under these conditions a low $KNO_3$ concentration, such as 0.25 mM, increased the early seedling growth of peas, but there was no effect on the germination rate. This effect was parallel with an increased water uptake by the seed. In that regard, water uptake is an integral requirement for the initiation and completion of the germination process [37]. In contrast, when 40 mM $KNO_3$ was applied directly to the dry seed, there was a significance decrease in water uptake. However, when the application of low $KNO_3$ followed imbibition in distilled $H_2O$ for 24 h, a contrasting effect on seedling growth was observed [12]. A minor effect was found at a low concentration (1 mM), whereas increased seedling FW and length were observed at a high concentration (30 mM). On the other hand, direct imbibition in 30 mM $KNO_3$ reduced both the fresh mass and length of seedlings [12].

### 4.1. Antioxidant Metabolism

There is limited information on the effect of $KNO_3$ priming on the antioxidant metabolism of plant seedlings. Only a few papers have reported some connection with POX, SOD, CAT and AOX enzymes [11,12]. In the present study, pea seedlings were found to contain very low DHAR activity, suggesting they mainly use MDHAR activity for ascorbate recycling, which utilises NADH as an electron donor. From an energy point of view, this is much more efficient than the DHAR pathway, which uses GSH as a source of reducing power [38]. Similarly, using the same pea cultivar, a higher MDHAR activity than that of DHAR was also reported at subcellular level [39,40]. Moreover, in pea seeds (cv. Alaska) imbibed in 20 mM $H_2O_2$, no DHAR activity was recorded, leading to a DHA accumulation in pea seedlings [6]. In addition, the seedlings subjected to 0.25 mM $KNO_3$ treatment displayed 2-fold higher MDHAR activity than seedlings treated with 40 mM $KNO_3$, suggesting a higher capacity to recycle ascorbate. A similar response was observed with APX and POX activity, indicating a reduced ability to control $H_2O_2$ levels in 40 mM $KNO_3$-treated seedlings when compared to controls and seedlings treated with a low $KNO_3$ concentration. MDHAR activity showed a different response to $KNO_3$ treatments in comparison to that which occurred with APX and POX activity. In this sense, low $KNO_3$ treatment increased both MDHAR activity and the $H_2O_2$-scavenging enzymes APX and POX, whereas high $KNO_3$ treatment produced a decrease in the activity of these enzymatic antioxidants. Both APX and some type of POXs can use ASC as an electron donor. These types of POXs can thus oxidise ASC and organic phenols at comparable rates [41]. Therefore, if APX and POX decreased, a lower level of ASC is oxidised and thus lower MDHA can be generated. This could explain the decline in MDHAR recorded by 40 mM $KNO_3$ treatment.

Similarly, $H_2O_2$-primed pea seeds also showed an increase in APX and POX activity [6]. It may be that pea seedlings treated with 0.25 mM $KNO_3$ displayed an efficient regulation of ROS generation when compared to those treated with 40 mM $KNO_3$, which in turn correlated with an enhanced seedling growth. The increase of POX and the maintenance of APX activity by $KNO_3$ treatment were also reported by Vidal et al. [12]. In tomato seeds, $KNO_3$ increased germination rates and enhanced SOD and catalase activities but had no effect on APX activity [11].

Information about the effect of $KNO_3$ treatments on the ascorbate pool is very scarce. Under the assayed conditions, $KNO_3$ progressively decreased ASC content. However, DHA only accumulated in seedlings treated with 40 mM $KNO_3$. According to our results, this response suggests that an effect on the biosynthetic pathway of ascorbate could take place. These results are in contrast to those reported by Vidal et al. [12], who observed a 3-fold increase in ASC in response to 10 mM $KNO_3$ or 50 μM sodium nitroprusside (SNP, a NO-donor). This may reflect the differences in the experimental procedures between the present study and the one reported previously [12]. Regarding glutathione levels, 0.25 mM $KNO_3$ produced a decrease in both GSH and GSSG, leading to a slight increase in the redox state of the glutathione pool. Likewise, 20 mM $H_2O_2$-treated pea seeds showed a decline in GSH and GSSG [6]. In this study, 40 mM $KNO_3$ had no effect on GSH but

reduced GSSG levels, thus increasing the redox state of the glutathione pool. The reduced accumulation in GSSG in both $KNO_3$ treatments can be linked to the observed increase in GR activity. Again, these results are in contrast to those reported by Vidal et al. [12], where no significant differences in GSH content with 10 mM $KNO_3$ were reported. In addition, these authors reported an accumulation of GSSG, which resulted in a decrease in the redox state of the glutathione pool. This highlights that the effect of $KNO_3$ treatment is highly dependent on the mode of application. Alternatively, decreased ASC and GSH levels in seed treated with 0.25 mM $KNO_3$ can be related to altered metabolism rather than reduced recycling of both molecules. In this sense, MDHAR, as well as DHAR and GR, activity was nearly 60% higher than in the control seedlings.

*4.2. Plant Hormones*

Potassium nitrate has been reported to affect GAs and ABA metabolism, though there is limited information available in this regard. It has been reported that exogenous $KNO_3$ or NO accumulation can modulate ABA and GAs content by increasing ABA catabolism as well as the up-regulation of GAs biosynthesis, respectively [15,18,20]. Vidal et al. [12] observed that $KNO_3$ or SNP treatments increased GAs levels and decreased ABA concentration in pea seedlings. These effects were reversed by incubation with cPTIO (a NO-scavenger), suggesting that, in part, the effect of $KNO_3$ in seedling growth and plant hormone levels may be due to partial generation of NO from $KNO_3$ [21]. In that sense, NO has a significant role in seed germination and the breaking of seed dormancy [9,22,27].

It is well known that GAs, together with ABA, are involved in seed dormancy and germination, promoting germination in many plant species [21,42]. In this study, a major proportion of GAs were present in the seedlings ($GA_1$ and $GA_4$), with only a small amount of $GA_4$ reported in the cotyledon. This suggests that during the growth and development of the seedling, GAs could be mobilised from the cotyledon to the rest of the plant, although an increase in GAs biosynthesis cannot be ruled out [20]. Hormone levels in particular organelles would seem to be dependent on metabolism and transportation [21]. GAs biosynthesis genes are expressed in different tissues in the embryonic axes of Arabidopsis seeds [42]. Therefore, it is likely that GAs and their precursors are actively transported inside the embryo and/or from the cotyledon to the embryo by specific transporters.

Decreased ABA levels in seedlings treated with 0.25 mM $KNO_3$ were correlated with enhanced seedling biomass. In addition, seeds treated with 40 mM $KNO_3$, which achieved the lowest seedling growth, also had the highest ABA levels in their seedlings. During seed dormancy breaking and germination in Arabidopsis, both a decline in ABA content and an accumulation of NO was reported [18]. This response was parallel to the induction of the *ABA-8-hydroxylase* gene, involved in ABA catabolism [15,18]. A decline in ABA levels has also been described in $H_2O_2$-treated pea seeds [6,8,43]. These authors suggested that this response can result from either the stimulation of ABA catabolism or the inhibition or slowdown of its biosynthesis.

As a result of the effect of $KNO_3$ on GAs and ABA content, a decrease in the ABA/GAs ratio was observed by 0.25 mM $KNO_3$, mainly in the seedlings, when compared to controls and samples treated with 40 mM $KNO_3$. It has been demonstrated that the ABA/GAs balance is crucial during the early stages of germination [44]. It has also been suggested that the key role of phytohormones such as ABA and GA during seed germination is interdependent with ROS metabolism [27]. A tight control of ROS production appears to be crucial for seed germination [45]. In fact, an accumulation of ROS and NO in germinating seeds, as well as an enhanced seed germination by the exogenous application of $H_2O_2$, has been reported [6,32,43]. Thus, $KNO_3$ can trigger a higher seedling growth by regulating ABA and GAs metabolisms, likely associated with an over-generation of NO from $KNO_3$.

## 5. Conclusions

The application of a low $KNO_3$ concentration to dry pea seeds promoted early seedling growth, which was linked to the maintenance (APX, SOD, POX) and/or the increase

(MDHAR, DHAR, GR) of antioxidant defences, leading to an efficient regulation of the ROS generation. In addition, and although 0.25 mM $KNO_3$ decreased GSH levels, a slight increase in the redox state of glutathione pool, which could be associated with an increase in GR, was observed. In general, a low $KNO_3$ concentration was associated with an increased $GA_1$ and decreased ABA in seedlings, which resulted in a decline in the ABA/GAs ratio. Data also suggest a modulation of GAs and ABA metabolism by $KNO_3$, in which a partial role of NO could not be ruled out. Furthermore, we suggest an interaction among $KNO_3$, antioxidant defences and the modulation of the ABA/GA ratio during the early growth of pea seedlings (Figure 5).

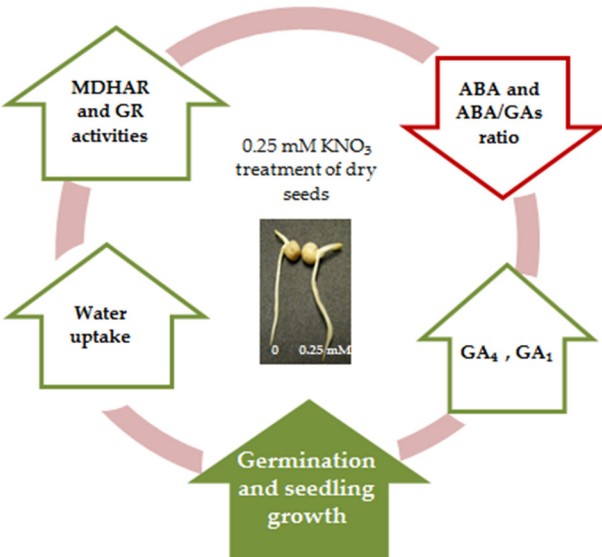

**Figure 5.** Simplified model summarising the effect of low $KNO_3$ concentration on promoting pea seed germination and seedling growth.

**Supplementary Materials:** The following are available online at https://www.mdpi.com/article/10.3390/seeds1010002/s1, Table S1a: Effect of different $KNO_3$ treatments on the germination rate of pea seeds, Table S1b: Effect of different $KNO_3$ concentrations on some growth parameters in pea seedling after 4 days of treatments.

**Author Contributions:** Conceptualization, J.A.H. and P.D.-V.; methodology, J.A.H. and P.D.-V.; validation, J.A.H., P.D.-V., G.B.-E. and J.R.A.-M.; formal analysis, J.A.H. and J.R.A.-M.; investigation, J.A.H. and P.D.-V.; resources, J.A.H. and P.D.-V.; data curation, J.A.H., P.D.-V., G.B.-E. and J.R.A.-M.; writing—original draft preparation, J.A.H. and P.D.-V.; writing—review and editing, J.A.H., P.D.-V., G.B.-E. and J.R.A.-M. All authors have read and agreed to the published version of the manuscript.

**Funding:** This research received no external funding.

**Institutional Review Board Statement:** Not applicable.

**Informed Consent Statement:** Not applicable.

**Data Availability Statement:** The study did not report any data.

**Acknowledgments:** We thank Isabel Lopez-Diaz and Esther Carrera for the hormone quantification carried out at the Plant Hormone Quantification Service, IBMCP, Valencia, Spain.

**Conflicts of Interest:** The authors declare no conflict of interest.

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
