# Peer review of "Potassium Nitrate Treatment Is Associated with Modulation of Seed Water Uptake, Antioxidative Metabolism and Phytohormone Levels of Pea Seedlings"

_2674-1024, doi:10.3390/seeds1010002_

Round 1

Reviewer 1 Report

The introduction should provide wider information on seed priming. It is too much focused on KNO3 in the current version. Several references (especially those arising from the authors #19-#26) are quoted in the introduction with no real explanations (page 2 lines 75-77 : The scientific literature contains plenty of information concerning the beneficial effects of ROS on germination and seedling growth processes [19–26]). This leads to a kind of over self-citations (23%). At the end of the introduction, it is not really clear why it is important to investigate the effect of seed incubation in different concentrations of KNO3.

English language needs to be improved and corrected.

Using the term of priming in the title of the present paper is highly confusing here since the authors proceeded to imbibition of pea seeds in the presence of difference concentrations of KNO3 which is far from priming. Indeed, seed priming is a pre-sowing treatment that improves germination rate, uniformity and seedling vigor. Pregermination metabolism is initiated using empirically determined imbibition parameters, then seeds are dried before they germinate. This is absolutely not the case here.

It is a pity that the results from the initial screening with a huge panel of KNO3 concentrations are not shown in the paper. Indeed, it will support the choice of the three targeted concentrations (0, 0.25, 40 mM KNO3).

In Figure 2, why is the dry weight not considered instead of the FW ? also in Figure 2, it is not clear which stage is considered (the authors must add this information to reconnect with Figure 1).

The figures for ABA and GA content are harldy understandable and must be improved (color code, presentation…).

The overall results suffer from a poor presentation and it seems that additional results (for instance concerning dehydration after KNO3 imbibition) are not presented here but elsewhere. The paper is of poor interest in its present form and I do not recommend its publication unless the comments above have been addressed.

To end, it would be helpful if the authors will summarize their results in a model.

Author Response

The introduction should provide wider information on seed priming. It is too much focused on KNO3 in the current version. Several references (especially those arising from the authors #19-#26) are quoted in the introduction with no real explanations (page 2 lines 75-77 : The scientific literature contains plenty of information concerning the beneficial effects of ROS on germination and seedling growth processes [19–26]). This leads to a kind of over self-citations (23%).

Authors: In response to the Reviewer, the information in this section regarding seed priming has been extended.

At the end of the introduction, it is not really clear why it is important to investigate the effect of seed incubation in different concentrations of KNO3.

Authors: Likewise, the motivation for the use of KNO3 has been further clarified. We tried to study if the direct nitrate addition could have any effect in the germination of pea seeds and in the seedling establishment and vigor, in which an interplay ROS/plant hormones/antioxidant metabolism may take place.

English language needs to be improved and corrected.

Authors:  English language has been carefully revised and numerous corrections have been introduced throughout the text in this regard.

Using the term of priming in the title of the present paper is highly confusing here since the authors proceeded to imbibition of pea seeds in the presence of difference concentrations of KNO3 which is far from priming. Indeed, seed priming is a pre-sowing treatment that improves germination rate, uniformity and seedling vigor. Pregermination metabolism is initiated using empirically determined imbibition parameters, then seeds are dried before they germinate. This is absolutely not the case here.

Authors: The point of the Reviewer is very relevant. We agree that priming sensu stricto involves re-drying of seeds after treatment. In our case, treatment and germination run at the same time, and the common goal of improving the germination rate, uniformity and vigor of the seedlings was intended. However, in order to agree with the priming definition, the introduction has been modified and manuscript title changed to refer the addition of KNO3 as a treatment during imbibition.

It is a pity that the results from the initial screening with a huge panel of KNO3 concentrations are not shown in the paper. Indeed, it will support the choice of the three targeted concentrations (0, 0.25, 40 mM KNO3).

Authors: In response to the Reviewer’s suggestion, a Supplemental Table 1 has been added to the manuscript in which we show the effect of the different KNO3 concentrations referred on seed germination and seedling growth.

In Figure 2, why is the dry weight not considered instead of the FW ? also in Figure 2, it is not clear which stage is considered (the authors must add this information to reconnect with Figure 1).

Authors: In addition to be a variable widely measured for seedling growth and vigor, the use of FW instead of DW allowed us the obtaining of crude extracts in the same material for the analysis of antioxidants and hormones.

With respect to the second observation, In Figure 2, we considered the data after 4 days of incubation. We have added now this information in the Figure legend.

The figures for ABA and GA content are hardly understandable and must be improved (color code, presentation…).

Authors: We apologize for the inconvenience. We consider, however, that Figures are correctly represented. For example, in Figure 3 we selected different colors depending of the GA showed (GA1 or GA4), whereas in Figure 4, the ABA levels in the seedling display a different color to that of the ABA present in cotyledons. In addition, we think that it is better to present the GAs or ABA levels from seedlings and cotyledons in the same plot just to have an idea of the level present in each organ. Nevertheless, we have made some small amendments in the figure legends, we have added the stage of the seedling and cotyledons for the measurements and we have added the color symbols in Figure 4b (that was missing. We are hoping that now the figures are more understandable.

The overall results suffer from a poor presentation and it seems that additional results (for instance concerning dehydration after KNO3 imbibition) are not presented here but elsewhere. The paper is of poor interest in its present form and I do not recommend its publication unless the comments above have been addressed.

Authors: Although we think that Results and Figures are clearly exposed, in agreement with the referee some minor modifications have been added in order to improve the manuscript presentation. A different issue is the lack or not of additional results; as we stated in point 3, the KNO3 treatment is not considered as priming and then dehydratation after KNO3 imbibition is not a needed step. As we point out above the objective was to elucidate the effect of KNO3 in seed germination and seedling growth rather than testing it effect as priming agent.

We sincerely consider that the reviewer concerns have been appropriately revised in the manuscript, making it suitable for its consideration for publication.

To end, it would be helpful if the authors will summarize their results in a model.

Authors: Following reviewer suggestion, a simplified model has been elaborated to visually summarize the effect of KNO3 on promoting pea seed germination and seedling growth.

Reviewer 2 Report

The study investigated the effect of KNO3 priming on pea seedling growth. They reported that pea seed treatment with low KNO3 level promote early seedling growth, which was associated with water uptake, antioxidative ability and ABA/GA levels. In my view, the work was absent of novelty and seems to be a repeated study of previous work as mentioned in the abstract. In addition, the discussion are most descriptive and did not well discuss the reasons why low KNO3 treatment promote seedling growth while high KNO3 inhibit seedling growth. Moreover, the antioxidant enzymes and ABA and GA content were determined for 4 days of seedling. Since the seedling growth has been detected to be affected by KNO3 at 4 day, the effect could come from the days before this day. Therefore, a time series sampling could be more accurate to reflect the influence of this chemical on seedling growth.

Others

  • MDHAR activity was highest in seedling treated with 0.25 mM KNO3, but similar in seedling treated with 0 and 40 mM KNO3, why APX and POX activities were similar in seedling treated with 0 and 0.25 mM KNO3, and lowest in seedling treated with 40 mM KNO3. Therefore, MDHAR activity has different response to KNO3 treatment in comparison to APX and POX activity. The authors should correct their description in 4.1 section.
  • I can not observe any column in the column chart in Figs. 2-4.

Author Response

The study investigated the effect of KNO3 priming on pea seedling growth. They reported that pea seed treatment with low KNO3 level promote early seedling growth, which was associated with water uptake, antioxidative ability and ABA/GA levels. In my view, the work was absent of novelty and seems to be a repeated study of previous work as mentioned in the abstract. In addition, the discussion is most descriptive and did not well discuss the reasons why low KNO3 treatment promotes seedling growth while high KNO3 inhibit seedling growth. Moreover, the antioxidant enzymes and ABA and GA content were determined for 4 days of seedling. Since the seedling growth has been detected to be affected by KNO3 at 4 day, the effect could come from the days before this day. Therefore, a time series sampling could be more accurate to reflect the influence of this chemical on seedling growth.

Authors: We are in disagreement with Reviewer´s overall impression about the novelty of the results. This work is not a repetition of a previous study for several reasons; firstly, in the referred previous study the KNO3 treatment was not applied under the same conditions (KNO3 was applied after imbibition in water) and concentrations. In this regard, as it is well known, the timing and treatment conditions are of key importance leading to different effect on germination. In the previous study, this was reflected in a distinct physiological response with respect to that reflected in the current manuscript. We have shown by this manuscript and from Vidal et al (2018) that depending of the mode of application, KNO3 concentrations stimulating seed germination and seedling are different.

Regarding the time series sampling, we did it in relation to germination rate, but not for FW or length, which were only analysed after 4 days of treatment.   As the Reviewer suggested, the experiments could have been carried out with 3-days old seedlings. However, extending the incubation for an extra day allowed us to have enough plant material to be uses in the different assays..

Others

MDHAR activity was highest in seedling treated with 0.25 mM KNO3, but similar in seedling treated with 0 and 40 mM KNO3, why APX and POX activities were similar in seedling treated with 0 and 0.25 mM KNO3, and lowest in seedling treated with 40 mM KNO3. Therefore, MDHAR activity has different response to KNO3 treatment in comparison to APX and POX activity. The authors should correct their description in 4.1 section.

Authors: Both APX and some types of POXs can oxidase reduced ascorbate (ASC). These types of POXs can thus oxidase ASC and organic phenols at comparable rates (Kvaratskhelia et al. 1997).  Therefore, if APX and POX decreased lower level of ASC is oxidized and thus lower MDHA can be generated. Probably this can explain the decline in MDHAR in the 40 mM KNO3-treated seedlings. In addition, less ASC is available in 40 mM KNO3-trated seedlings, and the decline in MDHAR is also reflected in the observed oxidized ascorbate accumulation.  Regarding the ASC levels, a progressive decrease occurred; according to our results, this response suggested that an effect on the ascorbate biosynthetic pathway could take place by KNO3 treatment.

Nevertheless, the information about the effect of KNO3 treatments on the antioxidant system is very scarce, and further research is required to elucidate its interaction with the different components of the antioxidant system.

I can not observe any column in the column chart in Figs. 2-4.

We apologize for the inconvenience. There must have been an error when converting to the pdf version which we hope it is now solved in the new version.

Reviewer 3 Report

I am writing about " Potassium nitrate priming is associated to a modulation of seed 2 water uptake, antioxidative metabolism and phytohormone 3 levels of pea seedlings". The manuscript is well written and it considered biochemical and phytohormne changes as affected by Potassium nitrate priming. It needs a minor revision and I released my comments at the pdf file. 

Author Response

REF 3

I am writing about " Potassium nitrate priming is associated to a modulation of seed 2 water uptake, antioxidative metabolism and phytohormone 3 levels of pea seedlings". The manuscript is well written and it considered biochemical and phytohormne changes as affected by Potassium nitrate priming. It needs a minor revision and I released my comments at the pdf file.

It is not clear how was the seed priming? in what temperature and how long seeds primed?

Authors: This information is specified in Material and Methods section (25 °C and up to 4 days)

Did you measure water absorption after seed priming?

Authors: We analyzed water absorption during the seed priming. To avoid misinterpretation, this has been specified in the corresponding Methods’ section.

Why did you placed individual seeds in plastic cups? For seed priming? Germination test? Water uptake test?

Authors: All seeds were placed individually on plastic cups during priming (days 0 to 4). This allowed us to follow water absorption in individual seeds.

Why did not use equal number of Petri dishes?

Authors: The number of petri dishes depended on the material available. In this sense, the statistical analysis conducted are equally feasible independently of the different number of petri dishes utilized.

Did you transfer seedlings directly in a freezer? or seedling first freezed in liquid nitrogen?

Authors: Seedlings were frozen first in liquid nitrogen, and then stored at -80ºC, and as such it is now specified  in the text.

I suggest to show germination data even if they are not significant. It is important to know how many of seeds were germinated. (pag 4)

Authors: In response to the Reviewer’s suggestion, this information has been now included in the Supplemental Table 1.

Reporting water absorption in units of water absorption as percentage of wet or dry weight of seedling is better understood than micro-mole H2O/weight.

Authors: There was a mistake  in the units, we apologize  for that. It must be µL H2O/g DW instead of µmol H2O/g DW . The results are referred to the dry weight of the seeds. Thus, the significance of the data does not change in comparison with the data expressed as percentage.

F value is not presented.

Authors: We added the F value for the redox state of glutathione. For DHA it is not possible to have a F value. We detected DHA only in samples treated with 40 mM KNO3. With only a group of values it is not possible to provide a statistical analysis.

There are 6 markers on the figure and legend of the figure shown by two different colors, but they can not be distinguished in the figure. (FIG 3 and Fig 4).

Authors: We apologize for the inconvenience. There must have been an error when converting to the pdf version which we hope it is now solved.

Reviewer 4 Report

Title: Potassium nitrate priming is associated to a modulation of seed water uptake, antioxidative metabolism and phytohormone levels of pea seedlings

In this study, the authors evaluated the potential positive effects of various KNO3 concentrations on modulation of seed water uptake, antioxidant defense systems, and phytohormone levels of pea seeds.

In the present form, the measured parameters seem loosely related. Also, the objectives are not very convincing. Why did the authors choose to pretreat seed with KNO3, was there any issue (seed dormancy) with the seeds of pea variety used in this study?  Is it economical to pretreat seeds under no stress conditions? What would be it benefit for the agriculture production system?

In title, to a… should be replaced “with”

For often, studying and/or regulating antioxidants are more preferable under stress conditions. Even, if the authors’ measured, what were the relations of these antioxidants with seed germination?

For all the parameters measured (like hormones and antioxidants), I assume the ultimate objectives were to enhance germination and seedling growth. However, some important measurements e.g. how KNO3 will affect the germination kinetics (germination percentage/potential, germination index, seedling vigor index etc.) are missing.

I would suggest the authors read and cite some of the most recent literature that would help in improving their manuscript

Kamran, M., Wang, D., Xie, K., Lu, Y., Shi, C., EL Sabagh, A., Gu, W., Xu, P., 2021. Pre-sowing seed treatment with kinetin and calcium mitigates salt induced inhibition of seed germination and seedling growth of choysum (Brassica rapa var. parachinensis). Ecotoxicol. Environ. Saf. 227, 112921.

Ren, Y., Wang, W., He, J., Zhang, L., Wei, Y., Yang, M., 2020. Nitric oxide alleviates salt stress in seed germination and early seedling growth of pakchoi (Brassica chinensis L.) by enhancing physiological and biochemical parameters. Ecotoxicol. Environ. Saf. 187, 109785.

Also, there is a major issue with the figures. In the downloaded file, I can only see Error bars and data points are missing. The authors should crosscheck it.

In references, 9 out of 35 references are self citations from the authors.  

Author Response

Title: Potassium nitrate priming is associated to a modulation of seed water uptake, antioxidative metabolism and phytohormone levels of pea seedlings

In this study, the authors evaluated the potential positive effects of various KNO3 concentrations on modulation of seed water uptake, antioxidant defense systems, and phytohormone levels of pea seeds. In the present form, the measured parameters seem loosely related.

Authors: We disagree with the Reviewer. An interrelation ROS/antioxidants/hormones during the seed germination process has been very well reported in the literature. In this sense, in our study, we found correlations among these components in response to the KNO3 treatment. Nevertheless, we are aware that further research is needed to determine accurately associations among antioxidant systems and phytohormones metabolism.

Also, the objectives are not very convincing. Why did the authors choose to pretreat seed with KNO3, was there any issue (seed dormancy) with the seeds of pea variety used in this study?  Is it economical to pretreat seeds under no stress conditions? What would be it benefit for the agriculture production system?

Authors: We tried to elucidate if direct nitrate addition could have any effect in the germination of pea seed and the seedling establishment and vigor. We decided pretreat with KNO3 because the information about the role of this compound (as well as the concentration) on seed germination is very scarce. In addition, the information on the effect of KNO3 priming on the antioxidant metabolism of plant seedlings is very scarce.

The economic implications of an extended use of KNO3 for seed priming are beyond the objectives of this work. Nevertheless, it indicated that only low KNO3 concentration (0.25 mM) can be effective to vigorize pea seedlings, which could eventually be costless.

On the other hand, we have not investigated whether or not the plants derived from the pretreatment are resistant to any abiotic stress. That deserves additional research, which opens a door for future work.

In title, to a… should be replaced “with”

Authors: The suggested change has been done.

For often, studying and/or regulating antioxidants are more preferable under stress conditions. Even, if the authors’ measured, what were the relations of these antioxidants with seed germination?

Authors: According to the results, in pea we can suggest that the treatment with 0.25 mM KNO3 led to an efficient regulation of ROS generation when compared to 40 mM KNO3-treated seedlings, which in turn correlated to an enhanced seedling growth and vigor. 48 h after treatment, the germination rate was higher in seeds pre-treated with 0.25 mM KNO3 than in seeds treated with 40 mM.

For all the parameters measured (like hormones and antioxidants), I assume the ultimate objectives were to enhance germination and seedling growth. However, some important measurements e.g. how KNO3 will affect the germination kinetics (germination percentage/potential, germination index, seedling vigor index etc.) are missing.

Authors: The supplemental Table 1 shows how KNO3 affected the germination kinetic during the treatment. It is true that after 3 days all the treatments reached 100% germination, but at 48 h, KNO3 concentration, in the range of 10-80 mM showed a delayed germination rate. In response to the Reviewer, the seedling vigor index has been introduced in Supplemental Table 1.

I would suggest the authors read and cite some of the most recent literature that would help in improving their manuscript

Authors: We thank the Reviewer for its suggestion. We have cited more recent and relevant papers (including Ren et al. 2020). Also, there is a major issue with the figures. In the downloaded file, I can only see Error bars and data points are missing. The authors should crosscheck it.

Authors: We apologize for the inconvenience. There must have been an error when converting to the pdf version which we hope it is now solved.

In references, 9 out of 35 references are self-citations from the authors. 

Authors: This work deals with a very specific topic in which the authors have recognized experience. We consider that all the manuscript cites are pertinent and can help to the understanding of the manuscript. In addition, a third of the self-citations corresponds to the methodology for plant material extraction and enzyme analysis, a field of research in which the senior author have an expertise of over 30 years.

Round 2

Reviewer 2 Report

The authors did not take the comments seriously and only revised the manuscript in a written form according to some of the comments. Nevertheless, the key problem that antioxidant enzymes activities should be determined in a time series was not addressed. In my view, this is essential for the study and could be a key point of novelty.  

Author Response

The authors did not take the comments seriously and only revised the manuscript in a written form according to some of the comments. Nevertheless, the key problem that antioxidant enzymes activities should be determined in a time series was not addressed. In my view, this is essential for the study and could be a key point of novelty. 

We kindly disagree with the Reviewer’s comment. It is true that a time series analysis always provides more information about any process under study. However, such time series analysis is beyond the objectives of this study. Herein, we want to study the interaction between antioxidant enzymes and ABA and GAs in response to KNO3 at a particular time, in which a clear growth promoting effect was observed. Since the water uptake test was carried out up to 4 days after imbibition, and at that time the effect of KNO3 on seedlings growth was analyzed, we select the same point for both the antioxidant metabolism and the levels of ABA and GAs determination. The conclusions obtained from these determinations stand alone, without an imperative need of comparing with other time points.

Reviewer 4 Report

Page 1 Line 31: Remove "on salt‐affected soils".

Page 1 line 33-37: rephrase the sentence, make it short

Author Response

Page 1 Line 31: Remove "on salt‐affected soils".

Page 1 line 33-37: rephrase the sentence, make it short

The minor changes suggested by Reviewer have been done.

In this sense, seed priming, defined as a pre-sowing treatment which involves controlled hydration of seeds during first stage of germination but does not allow radical protrusion through the seed coat, has been widely applied at both field and nursery production level as well as to improve the germination rate and seedling growth under different stress conditions